# Intracellularly Released Cholesterol from Polymer-Based Delivery Systems Alters Cellular Responses to Pneumolysin and Promotes Cell Survival

**DOI:** 10.3390/metabo11120821

**Published:** 2021-11-30

**Authors:** Tobias Kammann, Jessica Hoff, Ilknur Yildirim, Blerina Shkodra, Tina Müller, Christine Weber, Markus H. Gräler, Ulrich A. Maus, James C. Paton, Mervyn Singer, Anja Traeger, Ulrich S. Schubert, Michael Bauer, Adrian T. Press

**Affiliations:** 1Center for Sepsis Control and Care, Jena University Hospital, Am Klinikum 1, 07747 Jena, Germany; kammann.to@gmail.com (T.K.); jessica.hoff@med.uni-jena.de (J.H.); tina.mueller2@med.uni-jena.de (T.M.); markus.graeler@med.uni-jena.de (M.H.G.); ulrich.schubert@uni-jena.de (U.S.S.); michael.bauer@med.uni-jena.de (M.B.); 2Nanophysiology Group, Department of Anesthesiology and Intensive Care Medicine, Jena University Hospital, Am Klinikum 1, 07747 Jena, Germany; 3Laboratory of Organic and Macromolecular Chemistry (IOMC), Friedrich Schiller University Jena, Humboldtstr. 10, 07743 Jena, Germany; ilknur.yildirim@uni-jena.de (I.Y.); blerina.shkodra-pula@uni-jena.de (B.S.); christine.weber@uni-jena.de (C.W.); anja.traeger@uni-jena.de (A.T.); 4Department of Anesthesiology and Intensive Care Medicine, Jena University Hospital, Am Klinikum 1, 07740 Jena, Germany; 5Center for Molecular Biomedicine (CMB), Jena University Hospital, Hans-Knöll-Str. 2, 07745 Jena, Germany; 6Jena Center for Soft Matter (JCSM), Friedrich Schiller University Jena, Philosophenweg 7, 07743 Jena, Germany; 7Division of Experimental Pneumology, Medical School Hannover (MHH), Feodor-Lynen-Str. 21, 30625 Hannover, Germany; maus.ulrich@mh-hannover.de; 8German Center for Lung Research, Partner Site BREATH (Biomedical Research in Endstage and Obstructive Lung Disease Hannover), Carl-Neuberg-Str. 1, 30625 Hannover, Germany; 9Research Centre for Infectious Diseases, Department of Molecular and Biomedical Science, School of Biological Sciences, University of Adelaide, Adelaide, SA 5005, Australia; james.paton@adelaide.edu.au; 10Centre for Intensive Care Medicine, Department of Medicine, Wolfson Institute for Biomedical Research, The Cruciform Building, University College London Hospital, Gower Street, London WC1E 6BT, UK; m.singer@ucl.ac.uk; 11Medical Faculty, Friedrich Schiller University Jena, Bachstr. 18, 07743 Jena, Germany

**Keywords:** drug delivery, cholesterol, nanoparticle, microparticle, pneumolysin, pneumonia

## Abstract

Cholesterol is highly abundant within all human body cells and modulates critical cellular functions related to cellular plasticity, metabolism, and survival. The cholesterol-binding toxin pneumolysin represents an essential virulence factor of *Streptococcus pneumoniae* in establishing pneumonia and other pneumococcal infections. Thus, cholesterol scavenging of pneumolysin is a promising strategy to reduce *S.* *pneumoniae* induced lung damage. There may also be a second cholesterol-dependent mechanism whereby pneumococcal infection and the presence of pneumolysin increase hepatic sterol biosynthesis. Here we investigated a library of polymer particles varying in size and composition that allow for the cellular delivery of cholesterol and their effects on cell survival mechanisms following pneumolysin exposure. Intracellular delivery of cholesterol by nanocarriers composed of Eudragit E100–PLGA rescued pneumolysin-induced alterations of lipid homeostasis and enhanced cell survival irrespective of neutralization of pneumolysin.

## 1. Introduction

Cholesterol is a steroid metabolite produced in small amounts by all nucleated cells of the human body. Hepatocytes are the leading producers of cholesterol and control sterol homeostasis by internalizing and secreting cholesterol in lipoproteins. Cholesterol is critical in modulating biomembrane fluidity and is required as a necessary substrate for synthesizing steroid hormones, bile acids, and vitamin D [1,2]. Some pathogens have developed cholesterol-dependent cytolysins (CDCs) to recognize cholesterol in plasma membranes to identify and attack eukaryotic cells. One prominent CDC is pneumolysin (PLY), which is released by all clinical isolates of *Streptococcus* (*S.*) *pneumoniae* through autolysis. PLY is a critical virulence factor of *S. pneumoniae* and empowers the pathogen to cause life-threatening infections such as pneumonia, meningitis, and sepsis [3]. Monomers of the toxin bind in a stoichiometric ratio of 1:1 to cholesterol, resulting in the assembly and formation of a multimeric transmembrane pore complex that causes a rapid loss of membrane integrity with altered cellular ion homeostasis and cell lysis [4]. PLY mediates cell damage within the lung, breaching the alveolar epithelial barrier and enables pneumococcal tissue invasion [5]. 

Patients suffering from severe community-acquired pneumonia are commonly treated with combination antibiotic therapy based on β-lactams and macrolides [6]. However, bactericidal antibiotics such as β-lactams also increases PLY release, thus possibly contributing to complications such as uncontrolled immune responses and lung edema. On the other hand, macrolides decrease bacterial protein biosynthesis and, accordingly, PLY production, and may reduce the severity of pneumococcal infection [7,8]. However, given increasing macrolide resistance [9], other strategies are being investigated that may neutralize the adverse effects of PLY [10]. 

Given that biological lipid complexes, mammalian lipoproteins, are recognized as versatile extracellular mediators of inflammation [11] and scavengers of CDCs [12,13], the application of artificial cholesterol complexes mimicking these native biomolecules appears reasonable. Furthermore, the use of cholesterol-containing liposomes has been proven beneficial in pneumococcal infection models [14]. As liposomes or polymeric particles can be administered by inhalation, they reach the primary site of pneumococcal infection, preventing the detrimental effects of PLY [10]. 

This “scavenging strategy” employs the biophysical interaction of pore-forming toxins with its “anchor molecule”, namely cholesterol. By decreasing the likelihood that cytolysins interact with host membrane cholesterol and instead interact with exogenously administered cholesterol, the formation of cell-destructive pores can be attenuated if not prevented. Thus, such a strategy might effectively facilitate the protection of the delicate alveolar epithelial barrier, thereby preventing bacterial dissemination and eventually improving the outcome of life-threatening pneumococcal infection [15].

We previously showed that the host responds to pneumococcal pneumonia with the induction of hepatic cholesterol biosynthesis [12]. This hepatic adaptation is dependent on expression of PLY by the pneumococcal strain. It supports the scavenging hypothesis as the liver distributes lipids throughout the body, supplying peripheral tissues with cholesterol in the form of lipoproteins [2]. Similarly, cholesterol delivery systems are designed to increase cholesterol intracellularly. While both natural and artificial circulating cholesterol complexes protect cells by neutralizing PLY, the impact of increasing intracellular cholesterol levels on the cellular susceptibility to PLY-dependent pore formation is less well understood. We thus decided to investigate cellular defense mechanisms against PLY in the presence of polymer-based particles of different sizes and compositions prepared by nano- (NP) or microprecipitation (MP), increasing intracellular cholesterol. 

## 2. Results

### 2.1. Characterization of Polymer Particles 

Polymers were selected according to the required properties of the particles, namely biocompatibility and toxicity. As a consequence, the polymers chosen were (i) poly(lactic-*co*-glycolic acid) (PLGA, RESOMER RG 502 H, Evonik Industries, Essen, Germany) alone and in combination with the methacrylate copolymer poly((2-*N,N*-dimethylaminoethyl)methacrylate-*co*-butyl methacrylate-*co*-methyl methacrylate) (EUDRAGIT E100, Evonik Industries) (E100–PLGA). 

In addition, cholesterol was attached covalently to polylactide (PLA). Its transport to the site of action in the form of a polymer-based formulation is thereby ensured. For that purpose, cholesterol’s 3β-hydroxyl-group was used to initiate the polymerization of l-lactide to result in a bio-labile ester bond between the PLA-α-end group and the cholesterol (PLA–Chol) [16,17]. The cholesterol could hence be split off from the carrier to become bioavailable inside cells. Detailed descriptions of the polymer synthesis and characterization results are found in the Appendix A. 

The formulated particles require surfactants that could cause biological side effects depending on their concentration. Therefore, we used polyoxazoline-based surfactants, known for their excellent biocompatibility [18].

Polymeric cholesterol formulations containing PLGA, E100–PLGA blends (40:60 *w*/*w*), and PLA–Chol were prepared by nano- or microprecipitation. The mean diameter of different cholesterol-loaded carriers after lyophilization and resuspension was in the range of 270 nm for [E100–PLGA](Chol), 160 nm for [PLGA](Chol), and 300 nm for [PLA–Chol](Chol) when we applied nanoprecipitation and 690 nm for [E100–PLGA](Chol), 580 nm for [PLGA](Chol), and above 1000 nm for [PLA–Chol](Chol) for particles prepared by microprecipitation. The cholesterol cargo (and as a terminal moiety in PLA–Chol formulated particles) considerably influenced the size and lyophilization (Appendix A). LDH release was <10% among all nanoparticles tested at concentrations from 1 to 50 µg mL^−1^ (Figure 1a). Thus, these nanoparticles were considered biocompatible. However, at the highest dose (200 µg mL^−1^) [E100–PLGA] nanoparticles caused a significantly higher release of LDH than particles formulated from PLGA alone (27.0% compared to 1.2% for cholesterol-containing particles and 23.6% to 1% for vehicles), highlighting an efficient membrane penetration due to the use of E100 (Figure 1b) in the formulation. Particles of larger size composed of PLGA or PLA–Chol showed minor LDH release at all doses independent of the cholesterol cargo. This was to be expected as the microformulated particles were too large for the preferred endosomal internalization pathways. Hence, particles > 500 nm were less likely to enter the cells associated with inferior LDH release than the more cell penetrative nanoparticles.

To test whether the different cholesterol-containing nano- and micro-formulations can counteract PLY toxicity, we chose a non-toxic dose of particles (50 µg mL^−1^) and analyzed PLY-induced plasma membrane permeabilization LDH release in the presence of the particles. Since the non-toxic polyoxazoline was used as a surfactant, the particles were used after production and solvent evaporation without removal of excessive surfactant by centrifugation or filtration. The cholesterol loading of the different formulations was kept constant at 4% (*w*/*w*).

### 2.2. Effects of the Cholesterol Loaded Nano- and Microcarrier on Pneumolysin Toxicity

PLY can be detected at sublytic concentrations from 0.85 to 180 ng mL^−1^ in the cerebrospinal fluid of *S. pneumoniae* infected patients [19,20], although a recent study also suggested the presence of a higher PLY dose of 1 mg mL^−1^ in patients suffering from pneumococcal meningitis [19]. Based on these reports, we chose a lytic concentration of 250 ng mL^−1^, which is in the range of dosages previously applied by various laboratories [21,22]. 

The pneumolysin β-pore traverses the plasma membrane with a pore diameter of 26 nm [23] This allows the efflux of catalytically active tetramers of cytoplasmic LDH with a diameter of 10 nm through the pore while also promoting membrane rupture as a consequence of osmotic swelling [24]. Thus, LDH release may be considered a marker of toxicity regardless of the reason for its cellular release since the pore-forming ability of PLY is closely associated with its actual toxicity. First, we investigated the effects of cholesterol-containing particles on PLY toxicity with this assay after 3 h of co-stimulation (Figure 1c). [E100–PLGA] nanoparticles containing cholesterol could reduce the PLY-induced cell toxicity by up to 89%; however, the same was not observed for the [E100–PLGA] microparticles. This protective effect is dependent on cholesterol cargo since the unloaded particles failed to protect cells from PLY-induced LDH release. PLA–Chol-based particles rescued cells from PLY toxicity as microparticles but not as nanoparticles, while PLGA-based microparticles ultimately could not counteract the toxic effects of PLY. Concerning the efficient inhibition of PLY toxicity, [E100–PLGA] nanoparticles revealed the highest benefit for cell survival. Further investigation of the PLY inhibition kinetics by [E100–PLGA] nanoparticles suggests that these nanoparticles only delay or temporarily block PLY toxicity (Figure 1d) but do not permanently neutralize the toxin, as seen in studies where PLY-scavenging strategies were used before stimulation [12,13]. 

### 2.3. Mechanism of Protection from Pneumolysin by Nanoformulated Cholesterol

The addition of 5 µg cholesterol per mL formulated in [E100–PLGA](Chol)_NP_ increased the intracellular cholesterol concentration by about twofold, the highest among the screened nanoformulations. The exact amount of cholesterol dissolved in methanol and incubated with HepG2 cells achieved a similar increase; however, the increase was already observed after 3 h while nanoformulated cholesterol raised the concentration, mainly after 3 to 6 h. Cholesterol complexed with MCD resulted in the most pronounced 2,8-fold intracellular cholesterol increase, while polymers devoid of cholesterol failed to raise cellular cholesterol, MCD alone reduced cholesterol levels in HepG2 (Figure 2a, Appendix A). Notably, MCD experiments were performed separately from experiments with nano- or microparticles: absolute cholesterol levels thus differed between experiments (Appendix A). Furthermore, HepG2 cells are derived from a hepatocellular carcinoma cell line, known to have an efficient sterol metabolism. The high dynamics changes in cholesterol concentrations reflect the cells’ ability to process and metabolize cholesterol rapidly [25].

In contrast, the liquid-chromatography mass spectrometry (LC-MS) method to quantify cholesterol exclusively detects non-processed, i.e., unesterified cholesterol, due to their difference in mass transition [26]. Among the screened microparticles, the addition of 5 µg cholesterol per mL formulated in [PLA–Chol](Chol)_MP_ resulted in the highest increase of cellular cholesterol (Figure 2a), again concurring with protection against PLY toxicity (Figure 1c). However, the cell-protective effect was temporary as long-term exposure to PLY (9 h) overcomes the protective effect of cholesterol supplementation (Figure 1d), indicating a cholesterol-dependent molecular mechanism involved in the survival and repair of PLY-induced cell injury, and PLY scavenging.

Sterol-staining by filipin III allows for visualization of the cholesterol distribution in HepG2 cells (Figure 2b). Incubation of cells for 1.5 h with [E100–PLGA](Chol) nanoparticles showed an intracellular delivery of cholesterol (Figure 2b) in line with previous reports of the fast release kinetics of these nanoparticles. The delivered cholesterol accumulated in the plasma membrane and vesicles, a closer analysis of these cholesterol-rich-vesicles revealed a significant increase in number upon treatment with [E100–PLGA](Chol) nanoparticles (Figure 2c). The number of filipin III positive vesicles has been validated to quantify cholesterol uptake and processing in HepG2 cells [27]. Increased numbers of filipin III positive vesicles here depict the ability of HepG2 cells to efficiently internalize the nanoparticles. Their mean Feret’s diameter of 1.20 ± 0.3 µm and proximity to the plasma membrane leads to the conclusion that this was due to endocytosis of E100–PLGA nanoparticles. Therefore, we treated HepG2 cells with Pitstop-2, an inhibitor of endocytosis [28], which partly abolished the cholesterol-dependent cell-protective effects of [E100–PLGA](Chol)_NP_ (Figure 2d). The resulting increase in PLY toxicity of [E100–PLGA](Chol)_NP_ treated HepG2 cells after PS-2 incubation was associated with a significant decrease in intracellular cholesterol level between both groups (mean ± SD: 260 ± 39.8 vs. 197 ± 47.9 pmol cholesterol/sample [E100–PLGA](Chol)_NP_ vs. [E100–PLGA](Chol)_NP_ + PS-2; Appendix A). These results suggest the cholesterol dependency of the protective mechanism and support our hypothesis that intracellular cholesterol levels contribute to cellular defense against PLY [29]. The reduced uptake of [E100–PLGA](Chol)_NP_ indicates an ATP-dependent uptake, such as endocytosis. It is known E100 particles [30] and particle blends of PLGA and cationic Eudragit [31] offer improved cellular uptake, supporting our data. Hence, the uptake and endosomal release of the drug were enhanced by co-formulated Eudragit and PLGA. In addition, the final nanoparticles were shown to have high biocompatibility [32]. 

Little is known about how intracellular cholesterol metabolism responds to the formation of β-pores by cholesterol-dependent cytolysins. Reports suggest activation of caspases and inflammasomes due to, e.g., ion imbalances following β-pore appearance, and these subsequently mediate the adaptation of cholesterol metabolism [33]. However, the interaction between cellular stress sensors and cholesterol biosynthesis remains poorly understood.

HepG2 cells, used as model hepatocytes, take up serum cholesterol (LDL-cholesterol, VLDL-cholesterol) to cover metabolic demands but are also equipped with a highly efficient machinery that can synthesize and supply cholesterol throughout the body. Indeed, in response to PLY stress, the gene expression of key regulators of cellular lipid homeostasis, e.g., *sterol regulatory element-binding protein* (*SREBP)–1* and *SREBP–2*, are down-regulated (Figure 3a). Notably, HepG2 cells were stimulated under serum starvation, which would be expected to reduce intracellular cholesterol levels and activate the SREBP pathway to compensate for the limited availability of cholesterol.

The transcription of genes regulated by SREBPs, such as *β-hydroxy-β-methyl-glutaryl-CoA-reductase* (*HMGCR*)*,* the rate-limiting enzyme in cholesterol biosynthesis, or ∆*-6-desaturase* (*D6D*), involved in the generation of unsaturated fatty acids, remained largely unaltered upon PLY exposure. In contrast, PLY promoted the activation of the SREBP pathway at the protein level (Figure 3b). Its cleavage characterizes SREBP activation into two fragments that translocate into the nucleus promoting transcription of, for example, genes related to lipid and sterol signaling. In the presence of [E100–PLGA](Chol)_NP_ and elevated cellular cholesterol contents, HepG2 cells were insensitive to PLY-induced activation of the SREBP pathway, indicating that the nanocarrier’s intracellular delivery rescues the increased need for cholesterol during a PLY membrane attack. Moreover, PLY stress reduced the cellular HMGCR level (Figure 3c), demonstrating how PLY strongly interferes with the cellular machinery necessary for cholesterol biosynthesis. Again, supplementation of cholesterol by polymer particles stabilizes a key component of cellular cholesterol homeostasis.

Both severity and prognosis of pneumococcal infection are highly associated with the action of PLY [34,35]. The inhalation of cholesterol-containing lipid-based nanocarriers effectively inactivates PLY and prevents its toxicity at the primary site of infection in the lung and respiratory epithelia, respectively [14,15]. Our data suggest that apart from its known mechanism of protection, cholesterol also interferes with PLY toxicity in an intracellular manner. Encapsulation of cholesterol in polymer nanoparticles enables intracellular delivery and mediation of the intracellular effects of cholesterol while not interfering with PLY binding before cell recognition by the toxin. 

## 3. Materials and Methods

### 3.1. Synthesis and Characterization of Cholesterol Terminated Polylactide (PLA–Chol) 

Detailed descriptions of the polymer synthesis and characterization results are found in the Supplementary Methods.

### 3.2. Synthesis and Characterization of Cholesterol-Containing Particles

In addition to the PLA–Chol, we used (i) poly(lactic-*co*-glycolic acid) (PLGA, RESOMER RG 502 H, Evonik Industries, Essen, Germany) and (ii) the methacrylate copolymer poly((2-*N,N*-dimethylaminoethyl)methacrylate-*co*-butyl methacrylate-*co*-methyl methacrylate) (EUDRAGIT E100, Evonik Industries) (E100–PLGA) combined with polyoxazoline-based surfactants for biocompatibility [18].

Cholesterol carriers were prepared using two different precipitation techniques. Cholesterol-loaded carriers prepared by nanoprecipitation are referred to as nanoparticles (NPs) regardless of their final diameter. They were prepared using a solvent evaporation method with 50 mg of polymer (either PLGA, E100–PLGA (40:60 [*w*:*w*]), or PLA–Chol) and 5 mg of cholesterol (drug) dissolved in 5 mL of acetone. The polymer-drug solution was dropped into a 15 mL aqueous solution of 0.5% poly(2-methyl-2-oxazoline) (PMeOx, DP of 100) under constant magnetic stirring at 750 rpm utilizing a syringe pump (49 mL h^−1^). The resulting suspension was stirred for 24 to 48 h to evaporate the acetone. If the final volume was reduced, it was finally adjusted to 15 mL using type-1 water. 

Cholesterol-loaded particles prepared by microprecipitation are labeled as microparticles (MPs). These were prepared with a slightly different formulation of the neat material by a method modified from the above-described nanoparticle preparation: for the polyester-based formulations (PLGA and PLA–Chol), the polymer (50 mg) and the drug (5 mg) were dissolved in 2.5 mL of acetone. Then, the polymer–drug solution was transferred to a glass vial, and the surfactant–water solution was added dropwise under constant magnetic stirring at 750 rpm to the polymer–drug solution.

Particle size was determined by dynamic light scattering (DLS) using a Zetasizer Nano ZS (Malvern Instruments, Herrenberg, Germany) operating with a laser beam at 633 nm and a scattering angle of 173.8° (5 × 30 s, 30 s equilibrated at 25 °C). Diluted samples were used for concentration-independent measurements. For more extended storage periods, the particles were lyophilized for 48 h and stored at −80 °C.

### 3.3. Cell Culture

The HepG2 hepatocellular carcinoma cell line [36] was cultured in Dulbecco’s Modified Eagle Medium/Nutrient Mixture F-12 (DMEM:F12) (Biochrom, Berlin, Germany) containing 10% fetal calf serum (FCS) (Thermo Fisher Scientific, Langenselbold, Germany), 1% penicillin, and 1% streptomycin (Merck Millipore, Biochrom, Berlin, Germany). The supplemented medium is called DMEM:F12+, while without additives, i.e., no serum or antibiotics, it is called DMEM:F12. For experiments, culture media were replaced by DMEM:F12 to exclude the presence of serum during stimulation since serum proteins dramatically reduce PLY activity.

### 3.4. Stimulations

Cells were stimulated with polymer particles dissolved in 5% glucose solution and diluted in DMEM:F12 (final glucose concentration < 0.3%) for indicated time points up to 9 h. Controls received an equal volume of 5% glucose solution (without nano- or microparticles). For complexation and stimulation of cholesterol with methyl-β-cyclodextrin (MCD), 1 g of MCD (Sigma Aldrich, Darmstadt, Germany) was dissolved in 10 mL of type-1 water. Then, 30 mg of cholesterol was dissolved in 400 µL of isopropanol/chloroform (2:1) (Carl Roth, Karlsruhe, Germany). The MCD solution was heated to 80 °C, and cholesterol was added dropwise while stirring the solution. The final solution contained 6.8 mmol L^−1^ of cholesterol and 70 mmol L^−1^ of MCD. Alternative cholesterol was dissolved in methanol (Carl Roth, Karlsruhe, Germany) to a final concentration of 20 mmol L^−1^. A total of 5 µg of cholesterol per mL was used in all experiments, regardless of its formulation. PLY was isolated previously in the laboratory of Dr James C. Paton from *S. pneumoniae* and stored in 50% glycerol at −20 °C until use. PLY was diluted to a final concentration of 250 ng mL^−1^ in DMEM:F12 immediately before use [37]. For co-incubation experiments, polymer particles were added to the cells 30 min before PLY treatment.

### 3.5. Toxicity

Cytoplasmic lactate dehydrogenase (LDH assay) was used as a surrogate marker of cell lysis and membrane integrity. The CytoTox96 Non-Radioactive Cytotoxicity Assay (Promega, Walldorf, Germany) was utilized following the manufacturer’s instructions, including positive (maximum lysis), negative (unstimulated), and background controls. Absorbance was recorded on an EnSpire Microplate Reader (PerkinElmer, Beaconsfield, UK) at 490 nm. The absorbance of background and negative controls were subtracted, and the data normalized to the maximal absorbance of the positive control for each experiment. Data are shown as mean ratio ± standard error of at least triplicate determinations from a minimum of three independently performed experiments. Significance was assessed by two-way ANOVA and a post-hoc Dunnett’s test (* *p* < 0.05, ** *p* < 0.01, *** *p* < 0.001).

### 3.6. Cholesterol Quantification

HepG2 cells were cultured for 24 h in a 96-well plate in DMEM:F12+ before the media was exchanged to serum-depleted DMEM:F12. Cells were then stimulated with polymer particles (50 µg mL^−1^) at the indicated time points. After stimulation, the medium was removed and the cells lysed in 200 µL of methanol (99.9%, Chromasolv LC-MS grade, Honeywell Riedel-de Haën, Seelze, Germany) supplemented with 1.5 µmol L^−1^ of ergosterol (E6510, Merck KGaA, Darmstadt, Germany) as an internal reference for cholesterol quantification. Supernatants were transferred to glass vials (Dr. R. Forche chromatography, Moers, Germany) and stored at −80 °C until analysis. At each time point, levels of unesterified cholesterol were measured. The cholesterol content analysis was performed using liquid chromatography coupled to triple quadrupole mass spectrometry (LC-MS/MS). The Elite LaChrom LC system (Hitachi Europe GmbH, Duesseldorf, Germany) was equipped with an L-7250 autosampler and Merck Peltier Sample Cooler, for L-7250 and L-2130 liquid chromatography, and a L-2300 column oven. Gradient chromatographic separation was performed on a Zorbax Extend-C18 column (2.1 × 50 mm) (Agilent Technologies, Santa Clara, CA, USA) with a particle size of 3.5 µm. The column was maintained at 50 °C. Mobile phase A consisted of 0.1% (*v*/*v*) formic acid in deionized H_2_O and mobile phase B of 100% methanol. The column was equilibrated in 10% B with a flow rate of 0.5 mL min^−1^. The mobile phase switched to 100% B after sample injection. The flow rate increased linearly from 0.5 mL min^−1^ at 5 min to 1.0 mL min^−1^ at 7 min and remained constant until 10 min. Subsequently, the mobile phase changed to 10% B, and the flow rate decreased linearly from 1.0 mL min^−1^ at 10 min to 0.5 mL min^−1^ at 10.5 min and remained constant until the end of the program at 11.3 min. Detection took place between 2 and 10 min. The injection volume per sample was 10 µL. Samples were cooled at 8 °C.

The QTrap triple quadrupole mass spectrometer (AB Sciex, Forster City, CA, USA) equipped with an APCI source operating in positive mode was used for the detection of cholesterol under the following source parameters: source temperature 450 °C, curtain gas 40, collision gas “low” ion spray voltage 5500, ion source gas_1_ 60, ion source gas_2_ 30. In addition, both quadrupoles Q1 and Q3 were operated at mass resolution “unit”. As a result, cholesterol (but not esterified cholesterol) was identified with an *m*/*z* of 369.581 to 161.200 and ergosterol with an *m*/*z* of 379.305 to 69.200 in MRM mode. Finally, the Analyst 1.6.2 (AB Sciex, Forster City, CA, USA) software quantified cholesterol based on an external standard curve.

### 3.7. Gene Expression Analysis

A total of 5 × 10^5^ HepG2 cells were seeded in each well of a 6-well plate (Sarstedt, Nümbrecht, Germany) and cultured for 48 h before stimulation with PLY (250 ng mL^−1^). RNA isolation followed the guanidinium thiocyanate–phenol–chloroform RNA extraction principle and was performed according to the manufacturer’s instructions (Direct-zol RNA MiniPrep Plus, Zymo Research, Freiburg i. Breisgau, Germany). 

RNA concentration and purity were assessed spectrophotometrically on a NanoDrop 2000c (Thermo Fisher Scientific, Langenselbold, Germany). cDNA was synthesized by reverse transcription from a 500 ng template using the RevertAid First Strand cDNA Synthesis Kit (Thermo Fisher Scientific, Langenselbold, Germany). A BRYT Green I-based quantitative PCR (GoTaq qPCR Master Mix, Promega, Germany) was then performed from 25 ng cDNA, and 0.5 µmol L^−1^ forward/reverse primer (Appendix A) on a Rotor-Gene (Qiagen, Hilden, Germany). The transcriptional response to treatment was analyzed by comparing the mean quantification cycle (Cq) value (n ≥ 3) of stimulation to an untreated control by the following Equation (1).
ΔCq = Cq_Control_ − Cq_Stimulation_
(1)

In the absence of stable internal control reference genes, cDNA copy numbers of a transcribed gene were assessed semi-quantitatively by comparing treatments. However, transcript abundance is not relatively comparable between different genes but only within each condition by this method.

### 3.8. Cholesterol Staining

HepG2 cells were cultured in 8-well tissue-culture treated chambered coverslips (Ibidi, Munich, Germany) before the experiment. Cells were washed with phosphate-buffered saline (PBS), stimulated for 1.5 h, then washed with Hank’s balanced salt solution (HBSS) and fixed in 4% paraformaldehyde (PFA) solution. After washing with PBS containing 1 mg mL^−1^ of glycine (Carl Roth, Karlsruhe, Germany), cells were stained with 50 µg mL^−1^ of filipin III (Sigma Aldrich, Taufkirchen, Germany) for 15 min while protected from light. Cells were washed three times with PBS in the dark and covered with mounting medium (ROTI Mount FluorCare, Carl Roth, Karlsruhe, Germany). Cells were immediately transferred to the confocal microscope (LSM 780, Zeiss, Jena, Germany), and filipin III fluorescence was stimulated by two-photon excitation using a Ti:sapphire laser (Chameleon Ultra, Coherent Inc., Göttingen, Germany). Exciting light (720 nm) was reflected on the sample by a main-beam splitter 690+ through an LD C-apochromat 63 × objective (numerical aperture (NA) = 1.15 water corrected M27, Zeiss) on the specimen. Emitted light was split by a BS-MP 355/690 + R and passed through a short-pass filter (SP 485) before detection at a non-descanned detector. The focus plane was adjusted to detect the outer membranes of HepG2 cell clusters. Ten images were recorded for each condition on the experiment day in each of three independently performed experiments. Image data were analyzed using the FIJI distribution of ImageJ software. Representative images are shown.

### 3.9. Western Blotting

A total of 5 × 10^5^ HepG2 cells were cultured in tissue-culture treated 6-well plates for 24 h in DMEM:F12+ within an incubator. The media was replaced by DMEM:F12 before 3 h stimulation with PLY, [E100–PLGA](Chol)_NP_, or combinations thereof. Protein was collected by cell lysis in a radio-immunoprecipitation assay (RIPA) buffer supplemented with protease (Halt, Thermo Fisher Scientific, Langenselbold, Germany) and phosphatase (PhosSTOP, Roche, Basel, Switzerland) inhibitors. Protein concentration was assessed by bicinchoninic acid (BCA) assay (Serva Electrophoresis GmbH, Heidelberg, Germany), and SDS-PAGE was performed with equal amounts of protein under denaturing and reducing conditions. Gel loading was controlled by Coomassie gel staining. Protein was transferred on a 0.45 µm PVDF membrane (Carl Roth, Karlsruhe, Germany), and blots were blocked in 5% non-fat milk in Tris-buffered saline supplemented with 0.1% Tween (TBS-T). Blots were incubated with primary rabbit IgG against the target protein (anti-SREBP–2, ab30682, anti-HMGCR, ab174830, Abcam, Shanghai, PR China) diluted in 5% bovine serum albumin in TBS-T. Protein levels were analyzed by detecting chemiluminescence of horseradish peroxidase conjugated donkey anti-rabbit IgG (Jackson ImmunoResearch Europe, Ely, UK) on a charge-coupled device (LAS4000, GE Healthcare Life Sciences, Amersham, UK). Western blot signal density was analyzed by ImageJ software (Version 1.52n, National Institute of Health, Bethesda, MD).
SREBP–2 activation was calculated according to formula (2)
SREBP–2 activation = nSREBP–2/(full-length SREBP–2 + nSREBP–2)(3)

At least three independently extracted protein lysates per stimulation were tested by Western blot. Results displayed a representative Western blot and showed a mean protein level or ratiometric SREBP–2 activation compared to the untreated control and standard deviation. An unedited visualization of all replicas is provided with Supplementary Information.

### 3.10. Statistical Analysis

Statistical analysis is provided, if not otherwise described, in the figure legends. In Figure 3, the normalization procedure of the Western blot signal intensities to the control limits the statistical analysis for parametric testing and the variance analysis (ANOVA). Therefore, a non-parametric Kruskal–Wallis ANOVA was performed on the normalized data to account for non-equal distribution between groups. While Kruskal–Wallis ANOVA shows a significant activation of SREBP-2 (*p* < 0.05), group-wise comparison by a post-hoc Dunn’s test yielded no significant result (Figure 3b). The same was performed for the HMGCR level. Here, the difference between groups was assessed by Kruskal–Wallis ANOVA (*p* < 0.05). In addition, a post-hoc Dunn’s test revealed a significant difference between PLY and [E100–PLGA](Chol)_NP_ + PLY stimulated cells (*p* = 0.03). Due to the small sample size usually associated with Western blotting and the semi-quantitative nature of the method, the non-parametric Kruskal–Wallis ANOVA results were not shown in the figure. 

## 4. Conclusions

PLY impedes sterol biosynthesis in affected cells. The controlled intracellular release of cholesterol from polymer-based carriers in HepG2 cells stabilizes critical components of cellular cholesterol homeostasis during a severe attack of pneumolysin in lytic doses as is seen in patients suffering from pneumococcal infection. The rapid delivery of cholesterol by the polymer-based carrier employed in this study acted as an emergency supply for intrinsic cell defense mechanisms and this was distinct from toxin neutralization by scavenging before any cell interaction. Direct pneumolysin capture, as previously described for inhaled liposomes, seems unlikely. The nature of these defense mechanisms is yet to be characterized but requires intracellular lipid resources such as cholesterol (Figure 4) since their supplementation increased cell survival significantly. Hence, this study highlights the relevance of cellular cholesterol as an alternate therapy to improve host cell resistance to CDCs.

## Figures and Tables

**Figure 1 metabolites-11-00821-f001:**
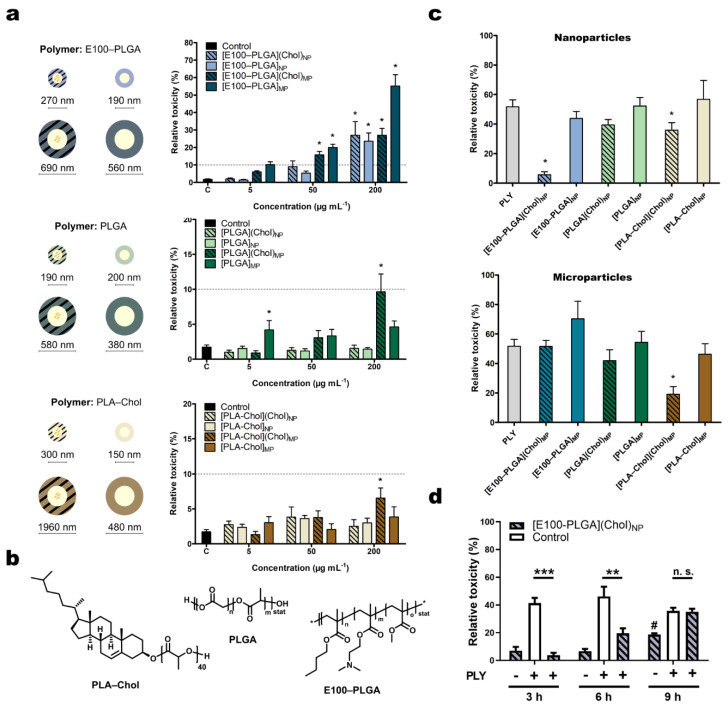
Evaluation of cell toxicity after polymer particle stimulation or pneumolysin stress. (**a**) HepG2 cells were treated with polymeric particles prepared by nano- or microprecipitation (NP or MP, respectively) of indicated concentrations in DMEM:F12 for 3 h. Stripes indicate the cholesterol cargo of the polymer particle. Mean diameter size (rounded) of tested sample and polymer composition (**b**) after lyophilization is shown. The polymer surface was coated with poly(2-oxazoline) (POx). (**c**) Cells were challenged with PLY (250 ng mL^−1^) for 3 h in the presence of nano- or microprecipitated particles (50 µg mL^−1^) before toxicity was examined by the release of cytoplasmic LDH. (**d**) Amelioration of PLY toxicity by [E100–PLGA](Chol)_NP_ was found to be time-dependent. Mean ± SEM toxicity is shown relative to completely lysed cells (= 100%). * *p* < 0.05, ** *p* < 0.01, *** *p* > 0.001, one-way ANOVA, corrected for multiple comparisons against untreated control (**a**) or PLY (**c**,**d**) (Dunnett’s test).

**Figure 2 metabolites-11-00821-f002:**
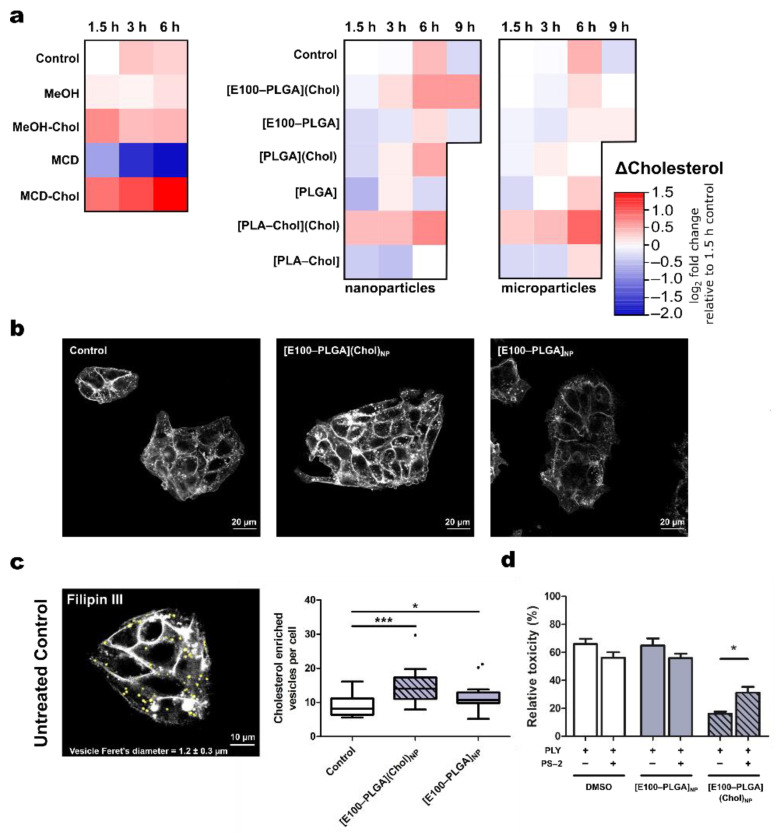
Intracellular delivery of cholesterol by polymeric particles. (**a**) HepG2 cells were treated with (50 µg mL^−1^) polymeric particles prepared by nano- and microprecipitation, or equimolar amounts of cholesterol dissolved in methanol (MeOH) or methyl-β-cyclodextrin (MCD) in DMEM:F12. The change in cellular un-esterified cholesterol is depicted as log_2_-fold relative to the cholesterol level of untreated control cells at 1.5 h; means and standard deviations are provided in Appendix A. (**b**) Distribution of cholesterol was assessed by fluorescence microscopy of filipin III stained HepG2 cells following 1.5 h stimulation with [E100–PLGA](Chol)_NP_ and [E100–PLGA]_NP_. The contrast has been adjusted equally on the depicted images for better visualization. Staining intensity does not reflect cholesterol levels as measured by mass spectrometry. (**c**) Cholesterol-rich vesicles (yellow dots) were identified in proximity to the plasma membrane by thresholding filipin III fluorescence images. The number of cholesterol-containing vesicles per cell was calculated after cell nuclei were counted. Feret’s diameter of the identified vesicles was examined by applying the analyzed particle function in the Fiji distribution of ImageJ. Supplementation of the nanocarrier significantly increases the number of cholesterol-rich vesicles per cell, as shown in Tukey’s box plot. (**d**) Inhibition of cellular uptake by 25 µmol L^−1^ PitStop-2 (PS-2) before 3 h stimulation with [E100–PLGA](Chol)_NP_ (50 µg mL^−1^) reduced protective effects of cholesterol supplementation as measured by LDH release assay. * *p* < 0.05, *** *p* < 0.001 tested by two-sided unpaired Student’s *t*-test.

**Figure 3 metabolites-11-00821-f003:**
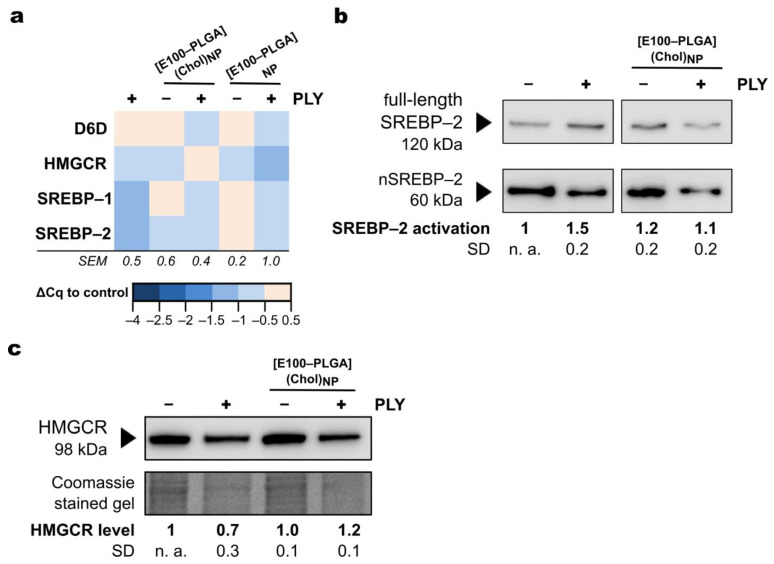
Intracellular delivery of cholesterol stabilizes critical components of the cholesterol biosynthesis pathway during PLY stress. (**a**) The cellular response to PLY (250 ng mL^−1^) was analyzed on a transcriptional level by semi-quantitative PCR in the presence of [E100–PLGA]-nanoparticles (50 µg mL^−1^). The level of transcriptional regulation was calculated from at least three replicated experiments and is shown color-coded compared to the expression of untreated control cells. SEM represents the mean inter-assay standard error of the genes listed under the indicated treatment. (**b**,**c**) Key regulators of cholesterol biosynthesis were immunoblotted after 3 h of treatment with PLY, cholesterol [E100–PLGA](Chol)_NP,_ or a combination thereof from at least three independently extracted protein lysates. One representative Western blot is shown. Blot images are cropped, and contrast has been adjusted for better visualization. An unedited full-size visualization of all replicas is provided with the Appendix A. (**b**) Activation ratio of SREBP–2 was calculated and is given in relation to the untreated control: SREBP–2 activation = nSREBP–2/(nSREBP–2 + full-length SREBP–2). (**c**) The level of detected HMGCR protein is shown relative to untreated control after correction of Western blot signal intensities by Coomassie gel staining.

**Figure 4 metabolites-11-00821-f004:**
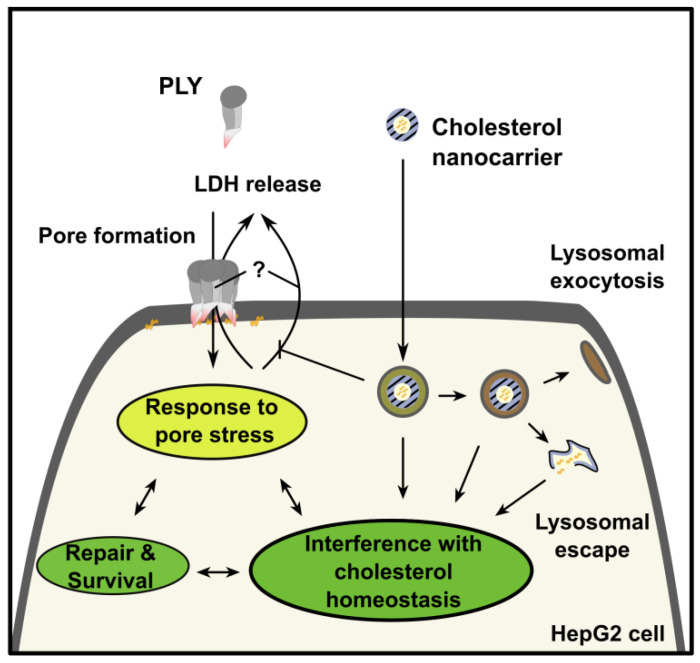
Interference with cholesterol homeostasis promotes cell-protective adaptation to pore formation. Intracellular cholesterol delivery promotes increased tolerance of HepG2 cells towards PLY pore-mediated plasma membrane injury and cell death. Supplementation of cholesterol by a nanocarrier such as [E100–PLGA](Chol)_NP_ prevented PLY-induced perturbation of the cellular signaling, and the activation of crucial lipid homeostasis regulators in response to PLY can enhance cell membrane repair.

## Data Availability

The figure data presented in this study are available in the Appendix A.

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
