# Peer review of "Intracellularly Released Cholesterol from Polymer-Based Delivery Systems Alters Cellular Responses to Pneumolysin and Promotes Cell Survival"

_metabolites, 2021, doi:10.3390/metabo11120821_

Round 1

Reviewer 1 Report

Authors:

The authors have addressed the concerns and performed additional experiments, which have improved the manuscript. Some points still need clarification.

Point 4: The authors now provide additional data (Figure 2a, left panel, and Supplementary Table 3) that include cholesterol delivery using cyclodextrins. The new data in Figure 2a shows the relative (-fold) change, but in the new Suppl. Table 3, the absolute/raw cholesterol levels in the control experiment using MCD is almost 3-fold lower than the control cholesterol levels in the nanoparticle and/or microparticle experiments. How comparable are the results if the control levels in the Nano-/microparticle experiments are even higher than the MBD-Chol loaded cells? This needs to be clarified.

Point 5: Rather than measuring cholesteryl esters, as requested by this reviewer, to prove bioavailability of the cholesterol taken up into cells via microparticles, the authors argue that measuring cholesterol is a reflection of the high dynamics of the cholesterol concentrations and the cells’ ability to process and metabolize cholesterol rapidly and quote ref [25]. The HepG2 cell line can effectively store cholesterol in lipid droplets (Hepatol Commun. 2019 Apr 1;3(6):776-791; J Lipid Res. 2017 Jun;58(6):1067-1079). Hence, evidence for the bioavailability of the particle-delivered cholesterol is still missing.

Point 7: Many researchers have successfully provided quantification of filipin staining in cell-based experiments. If new experiments are not performed, the mismatch of staining intensity and biochemical cholesterol measurements should be explained in the text.

Point 9: If ‘there is hardly a process that is not affected by this solid toxic stimulus and common "Housekeeping proteins" in these experiments are highly affected by the PLY treatment, how meaningful is the change in SREBP and HMGCoR levels/activation if as explained by the authors, events like membrane rupture occur, which would trigger major apoptotic events. If delivery of cholesterol reduces the PLY stress, this should also be reflected in the housekeeping genes and/or cell viability tests?

Reviewer 2 Report

General: Overall, I think the authors have made improvements to the manuscript and the additions they have made make the work stronger. I have no further substantial concerns, below are my responses to their replies.

Responses

  1. I thank the authors for their thoughtful responses to my question about generalizability to different cell types along the respiratory tract and potential uses outside of hepatocyte cholesterol metabolism. I agree that this is a novel intracellular mechanism that is independent of the “decoy strategy” that has been described and feel the authors do address the issue in discussion and conclusion. One unresolved question in my mind, is how might this technology be used upstream in pneumococcal pathogenesis, i.e. before pneumolysin has entered the circulation and influences hepatic cholesterol metabolism? I think it might be of interest to know how potential inhaled delivery might (or might not noting the limitations that the authors bring up in their comprehensive reply) be able to halt progression from upper respiratory tract colonization, to lower respiratory disease, to invasive disease. I feel there might be future benefit to understanding how cells along the respiratory tract (not just alveolar type I and II epithelial cells) might respond to the delivery of intracellular cholesterol. One such application in current clinical use is Amikacin Liposome Inhalation Suspension (ALIS) that has been shown to be able to be delivered to multiple lung tissue compartments (PMID 29867826).

  1. Thank you for the clarification in timing and think this is an important methodological addition. I understand the current limitations to the model and that this is a necessary first step towards investigation in more complex animal models.

  1. Agree that this would be an important area of future research given limitations of LDH to define mechanisms of cell death and membrane permeability.

  1. Thank you for addressing this point and I feel Supp Figure 3 is an informative addition to the manuscript.

  1. I appreciate this information. I might make this clearer in Figure 4 as it is depicted as LDH leaving the PLY-induced pore.

  1. Thank you for the clarification.

  1. I think it would be reasonable to include the non-parametric statistical analysis and list the limitations as you have in the response, which would be my bias.

Round 2

Reviewer 1 Report

The authors have discussed all points raised by this reviewer. Some of the responses provided, e.g. batch- or environment dependent 3-fold differences in background 'metabolic activities', or technical limitations when using filipin, appear weak, but changes in the manuscript are provided for the reader that address these points. The manuscript can be accepted in its present form.

This manuscript is a resubmission of an earlier submission. The following is a list of the peer review reports and author responses from that submission.

Round 1

Reviewer 1 Report

This is an interesting manuscript describing the generation of cholesterol-containing polymer particles and their delivery to cells, using the HepG2 cell line as a model. Results show some increase of cholesterol content in cells and alteration of SREBP and HMG-CoA reductase levels. From these findings the authors conclude that cholesterol homeostasis is altered and contributes to cellular defence mechanisms against pneumolysin (PLY). However, the findings are still preliminary and some major points need to be addressed to draw such conclusions.  

  1. The description of the polymer synthesis and their characterization in the first 3 paragraphs of the Material and Methods is written like a result section. While the data is presented in the Supplementary Methods, the authors should consider to re-organize the manuscript and move this section to the Results.
  2. While cholesterol measurements were performed from cells grown in 10% FCS-containing media (section 2.5), the samples for western blotting were prepared from cells incubated for 3h in serum-reduced media (section 2.8). Why are the methods for these two methods different? Serum depletion would be expected to reduce cellular cholesterol levels and/or downregulate basal signalling pathways, both possibly impacting on SREBP processing and/or HMG-CoAR levels?
  3. Figure 2a-b shows cholesterol levels from cells incubated +/- cholesterol-loaded particles. The data is shown as log2-fold change and after 6 – 9 h a maximal 2-fold increase in cellular cholesterol is observed. Shorter incubations (1.5, 3 h) only result in minor or no change in cellular cholesterol levels. This seems rather marginal compared to the ‘old-fashioned’ delivery of cholesterol using cyclodextrins, which can result in at least 2-fold increased cellular cholesterol after only 60 min (Christian et al., JLR 1997, 38, 2264-72, 1997; Blom et al., Biochemistry 2001, 40, 14635-44). The authors should consider to compare the cholesterol-loading efficacy with other, more established procedures. Moreover, this control experiment would allow to address a critical aspect of this manuscript: if cholesterol delivery independent of the newly described carriers here, and proven to be bioactive, could protect against PLY.
  4. Moreover, at these later time points (>2h), one would expect a substantial proportion of internalized cholesterol to be esterified. This activity occurs in the ER, where SREBP is also processed. Delivery of radiolabeled cholesterol incorporated into polymer particles would allow the authors to not only confirm cholesterol uptake, but also provide evidence for its bioavailability after endocytosis and transport into the ER.
  5. What are the cellular cholesterol levels when using the uptake inhibitor PitStop2? Filipin staining of cells incubated with this inhibitor would also be convincing: if membrane staining is reduced with Pitstop2, the authors could rule out passive diffusion into the cell membrane rather than endocytic processes as underlying mechanism. 
  6. Filipin staining of E100-PLGA in panel C is weaker than control, yet the quantification of cholesterol-enriched vesicles in panel 2D is similar for these two conditions. How is this possible?
  7. Figure 3: incubations with cholesterol-containing polymer shows less than 2-fold changes in HMG-CoAR and SREBP2 mRNAs, while SREBP2 activation increases 1.2 fold and HMG-CoAR protein levels remained unchanged. PLY incubations resulted in 1.5 fold SREBP activation and 0.3-fold decreased HMGcoAR levels. The authors interprete these findings as evidence for ‘strong interference’ with cholesterol biosynthesis and homeostasis. This is an over-interpretation and to support this hypothesis, at least radioactive experiments (incorporation of labeled acetate into cholesterol, endogenous cholesterol into cholesterol ester) are needed.
  8. Figure 3b-c: The authors should consider to include control blots showing actin, GAPDH (or similar) to their western blot analysis to exclude an overall reduction in protein synthesis/levels in the cell lysates obtained after PLY exposure (panel b) and improve the limited quality of the loading control (coomassie) panels in panel 3C.

Reviewer 2 Report

General: Kammann et al present a study analyzing the ability of intracellular cholesterol from nano- or microparticles to blunt the cytotoxicity of pneumolysin. My background is a clinical translational scientist so I will not comment on the synthesis and characterization of the polymers used. Though, I found this work important and novel in its approach to a fundamental issue in host-pathogen biology. My comments below concern the model for assessing cytotoxicity and some of the other putative pathways that would be of interest for both the NP and MP in their ability to blunt PLY injury. 

Major comments:
1. Cell type: stated in the introduction that liposomes targeting the primary sites of pneumococcal infection are an attractive therapy via the "scavenging strategy". HepG2 cells seem to be a reasonable choice to understand issues of cellular trafficking of cholesterol and the cell death dynamics given the hepatocytes primary role in cholesterol metabolism; however, as a clinical model it would seem respiratory epithelia might be a better choice. How well do the authors think the findings of the current manuscript would be generalizable to the respiratory epithelium -- which would likely be the primary source of contact with pneumococcus. Would there be higher toxicity of intracellular cholesterol? Differences in PLY injury and ability to blunt with NP/MP?
2. Timing of injury/particle delivery: I think it could be a bit clearer in the methods section (2.3) about when PLY and the polymers were applied to the cell cultures. Secondly, in pneumococcal infections the host is exposed to the bacterium and PLY before coming to clinical attention. I would be interested to know the efficacy of the [E100-PLGA](Chol)NP or the other constructs when applied after PLY exposure (as there can be some PLY before antibiotics because of bacterial autolysis).  
3. Apoptosis, necroptosis, and calcium influx: LDH release may not effectively capture the cellular stress signals from apoptosis (PMID: 27026501) and may not be able to effectively determine necrosis from necroptosis (PMID: 28387756). Additionally, sub-lytic effects of PLY can be important and seem to be mediated through intracellular calcium (PMID: 31914676). Could the authors comment on these alternative cell-death and stress pathways and how the NP/MP might influence these axes.
4. Intracellular cholesterol: Figure 2e suggests that a significant reduction in PLY toxicity in the setting of inhibition of endocytosis but there still appears to be an almost 50% reduction without. What mechanism do the authors propose explains this? 

Minor comments:
1. Figure 4: Is it known that LDH release through same pores? 
2. Figure 4: It does not seem currently clear how the polymers and cholesterol flux through the endolysosomal system. 
3. Fig S1 could be highlighted (line 111)
4. Statistical testing to support changes in figure 3?
